# An Algorithm for Learning Switched Linear Dynamics from Data

**Guillaume Berger**[*]    **Monal Narasimhamurthy**[*]    **Kandai Watanabe**
**Morteza Lahijanian**    **Sriram Sankaranarayanan**
University of Colorado Boulder, Boulder, CO, USA
`firstname.lastname@colorado.edu`

## Abstract

We present an algorithm for learning switched linear dynamical systems in discrete time from noisy observations of the system's full state or output. Switched linear systems use multiple linear dynamical modes to fit the data within some desired tolerance. They arise quite naturally in applications to robotics and cyberphysical systems. Learning switched systems from data is a NP-hard problem that is nearly identical to the $k$-linear regression problem of fitting $k > 1$ linear models to the data. A direct mixed-integer linear programming (MILP) approach yields time complexity that is exponential in the number of data points. In this paper, we modify the problem formulation to yield an algorithm that is linear in the size of the data while remaining exponential in the number of state variables and the desired number of modes. To do so, we combine classic ideas from the ellipsoidal method for solving convex optimization problems, and well-known oracle separation results in non-smooth optimization. We demonstrate our approach on a set of microbenchmarks and a few interesting real-world problems. Our evaluation suggests that the benefits of this algorithm can be made practical even against highly optimized off-the-shelf MILP solvers.

## 1   Introduction

Switched linear systems model dynamical systems that arise in diverse areas including natural sciences (biological models) [4, 8, 15], robotics [2], and cyber-physical systems [1]. They are characterized by a finite set of modes, wherein each mode features a different set of governing equations for the future states in terms of the current state of the system. In this paper, we study algorithms for inferring the dynamics of a switched linear system from data that consists of full-state observations (this can be subsequently extended to output observations using ARX models; see Section 2 for details). Specifically, given a set of data points $\{(\mathbf{x}_i, \mathbf{x}'_i)\}_{i=1}^N$ of current and next state observations, and a number of modes $m$, we wish to find matrices $A_1, \ldots, A_m$ such that each data point $(\mathbf{x}_i, \mathbf{x}'_i)$ is explained by some matrix $A_j$: $(\forall i) \ (\exists j) \ ||\mathbf{x}'_i - A_j\mathbf{x}_i|| \leq \epsilon ||\mathbf{x}_i|| + \tau$, for given error tolerances $\epsilon, \tau > 0$. However, the switching signal which governs the current mode associated with each data point is latent. This renders the problem NP-hard and the best known approaches have exponential complexity in terms of the number of data points and the desired number of modes [22]. The exponential complexity in the number of data points makes this problem especially challenging since many datasets of interest have thousands of data points but with tens of modes and state variables.

The key insight in this paper is that by reformulating this problem to incorporate a "gap" in the tolerance, we can significantly improve the complexity to be *linear* in the number of data points $N$ and exponential in the number of modes and the dimension of the state space. Rather than providing a YES/NO answer for a single tolerance bound $\epsilon$, our approach takes as inputs two levels of tolerance $\epsilon_1 < \epsilon_2$. It either finds matrices that satisfy the "upper" tolerance bound $\epsilon_2 > 0$ or

36th Conference on Neural Information Processing Systems (NeurIPS 2022).

returns No if no set of $m$ matrices can fit the data with the "lower" tolerance $\epsilon_1$ ($< \epsilon_2$). However, if the underlying data could have been modeled with some tolerance that lies in the gap $(\epsilon_1, \epsilon_2)$, our algorithm can either succeed or return No with no guarantees provided. Our technical approach exploits the gap in our formulation to argue that any set of solutions must have a lower bound on its volume. Furthermore, by exploring the solution space in a carefully defined manner, we guarantee that each step of our approach will shrink the volume of the remaining solution space by a constant factor. This is achieved by using ideas that are similar to separation oracles in combinatorial optimization [20], or the so-called "cutting plane" argument [10]. However, our approach applies these inside a tree-based branch-and-bound algorithm, wherein we use the volume contraction to prove bounds on the depth of the underlying tree. Thus, we achieve a bound on the running time that is exponential in the state dimension and the number of modes but linear in the data size.

We demonstrate our ideas using a prototype implementation that is compared against a standard mixed-integer linear programming (MILP) formulation of the problem solved using a state-of-the-art solver (Gurobi) [17]. Our experimental results confirm that the theoretical insights also apply in practice to yield an algorithm that is often orders of magnitude faster, especially as the amount of data increases. We also demonstrate our approach using two interesting applications that include modeling human writing on a tablet in order to predict what alphabet is being written from a small number of samples, and deriving models of mechanical systems with contact forces from data.

## 1.1 Related Work

The problem of switched and hybrid system identification has been widely studied using a variety of approaches going back to the early 1980s; see, e.g., [32]. We discuss a few representative approaches, and refer the reader to the monograph by Lauer et al [22] for further details.

The identification approach may be *exact* (seeking a global solution that minimizes the error between the data and the model prediction), or *approximate* (wherein the optimization problem is solved approximately, or assumptions about the nature of the switching signal are used to simplify the problem). Our approach here is *exact* but the problem itself is reformulated with a gap. As for the nature of the switching mechanism, many approaches (including ours) focus on identifying the dynamics of individual modes while assuming that the switching signal is *exogenous*; on the other hand, other approaches, such as *piecewise affine system identification* [5, 28], *hybrid automata learning* [29], or *linear complementary systems learning* [19], attempt to identify the dynamics of each mode along with rules for transitioning between modes.

Vidal et al [33] present an exact approach for finding matrices that fit the data in the absence of noise by posing the problem as one of finding zeros of a multivariate polynomial. This is subsequently refined by Ozay et al [26] to handle the case of noisy data, by using sum-of-squares relaxations to obtain a semidefinite optimization problem. Assumptions on the nature of the switching signal can drastically simplify the problem at hand. For instance, Ozay [25] shows that the problem can be solved in polynomial time using a dynamic programming algorithm if the number of mode switches in a trajectory is bounded. Other approximate approaches involve greedy algorithms [5] and block–coordinate descent (similar to $k$-mean regression) algorithms [21, 31]. However, the performance of these algorithms can vary depending on the nature of the data and no guarantees are available.

Machine learning techniques have also proved useful in hybrid system identification. For instance, standard machine learning approaches (such neural networks) can be used to infer hybrid system models [16, 24]. However, the resulting number of modes can be exponential in the network size. Moreover, switched systems often involve discontinuous switching which cannot be modeled adequately using standard activation functions such as sigmoids or ReLU. Ly and Lipson present an approach for learning hybrid automata from data using evolutionary techniques [23]. Their approach combines symbolic regression for learning the dynamics of the individual modes with techniques for guessing the latent modes and inferring the conditions for switching from one mode to another.

Also relevant to this work, Dempster et al [12] study the problem of inferring hidden Markov models with linear systems, called *Jump Markov Models* (JMMs), using a modification of the *Expectation–Maximization* (EM) algorithm. However, the E-step, which infers the assignments of modes, can be quite expensive for JMMs. Subsequently, Gharamani and Hinton [14] propose a technique wherein the difficulties in the E-step are resolved using variational inference, whereas Blake et al [7] propose a sampling-based scheme. A key difference between our approach and JMMs is that we focus on

the algorithmic efficiency for learning the dynamics of the modes. Although this requires us to infer the sequence of modes for the data, we do not learn the Markov model that generates this sequence. However, our example on handwriting recognition presented in Section 4 illustrates how off-the-shelf approaches for learning a finite-state model can be combined with our approach in a fruitful manner. As for our application involving mechanical systems with contact forces (Section 4), a recent work by Jin et al [19] propose a convex relaxation approach to learn *linear complementary systems*. These systems provide compact representations for a class of piecewise linear systems, including mechanical systems with contact forces [3, 27]. Our implementation does not use this compact representation, but is nevertheless able to learn switched linear models for these systems. Finally, our approach side-steps the problem of estimating the number of modes which is often an issue. This problem can be addressed by adding a penalty term accounting for the "model complexity" to the overall objective function to be minimized [23] or by using more sophisticated non-parametric Bayesian approaches wherein the prior distribution specifies unbounded number of modes and the varying number of model parameters varying as the number of modes increases [13].

## 2 Problem Formulation

We will use bold, lower case symbols $(\mathbf{x}, \mathbf{y}, \mathbf{w})$ to denote vectors, and upper case letters $(A, B, C)$ to denote matrices, unless otherwise mentioned. Let $\mathbb{R}$ denote the set of real numbers, and $\mathbb{N}$ the set of natural numbers. For $m \in \mathbb{N}$, let $[m] = \{1, \ldots, m\}$. Let $E_n$ denote the $n \times n$ matrix with all entries equal to 1. We write $A \leq B$ to denote that each entry of $A$ is less than or equal to the corresponding entry of $B$ (same notation will be used for vectors).

A linear dynamical system in discrete time is defined by its state $\mathbf{x}(t) \in \mathbb{R}^n$ and output $\mathbf{y}(t) \in \mathbb{R}^p$, at each time $t \in \mathbb{N}$. The state evolves using the dynamical rule: $\mathbf{x}(t+1) = A\mathbf{x}(t)$, wherein $A \in \mathbb{R}^{n \times n}$. The output is obtained as $\mathbf{y}(t) = C\mathbf{x}(t)$, wherein $C \in \mathbb{R}^{p \times n}$.

**Definition 1** (Switched Linear Dynamical System)**.** A switched linear dynamical system with state $\mathbf{x} : \mathbb{N} \to \mathbb{R}^n$ and output $\mathbf{y} : \mathbb{N} \to \mathbb{R}^p$ with $m \geq 1$ modes is defined by $m$ matrices $A_1, \ldots, A_m$. For each time $t \in \mathbb{N}$, there exists a mode $\sigma(t) \in [m]$ s.t. $\mathbf{x}(t + 1) = A_{\sigma(t)}\mathbf{x}(t)$. The *switching signal* $\sigma : \mathbb{N} \to [m]$ chooses a mode for each time $t$ such that the continuous state $\mathbf{x}(t)$ evolves according to this mode. The output is given by $\mathbf{y}(t) = C\mathbf{x}(t)$ according to a given matrix $C \in \mathbb{R}^{p \times n}$.

*Remark* 1. Switched *affine* systems, with $\mathbf{x}(t + 1) = A_j\mathbf{x}(t) + \mathbf{b}_j$ for $\mathbf{b}_j \in \mathbb{R}^n$, are handled by augmenting $\mathbf{x}$ with an extra component that is always 1, thus treating $\mathbf{b}_j$ as a new column of $A_j$.

The goal of this paper is to learn a switched linear dynamical system model from observation data. The simplest type of data consists of full state observations involving pairs of states of the form $\{(\mathbf{x}_i, \mathbf{x}'_i)\}_{i=1}^N$, wherein each $\mathbf{x}_i = \mathbf{x}(t_i)$ and $\mathbf{x}'_i = \mathbf{x}(t_i + 1)$ are states observed at two successive time instants $t_i$ and $t_i + 1$. In particular, the switching signal $\sigma$ is *not* observed. We assume that the number of modes $m$ is known.[1]

*Remark* 2. The assumption of full state observation is often impractical. However, our techniques extend to output observations by using a fixed-length history of output observations $\mathbf{y}(t - p), \ldots, \mathbf{y}(t)$ and finding a switched auto-regressive (SARX) model [35] of the form:

$$\mathbf{y}(t + 1) = \sum_{k=0}^{p} B_{\sigma(t),k}\mathbf{y}(t - k), \text{ wherein } \sigma(t) \in [m], \ B_{j,k} \in \mathbb{R}^{p \times p}.$$

Therefore, in the following, we will focus our description on the full-state data.

**Definition 2** (Fitting With Error Bound $\tau$ and Tolerance $\epsilon$)**.** A set of matrices $A_1, \ldots, A_m$ "fits" the data $D$ with *error bound* $\tau > 0$ and *tolerance* $\epsilon > 0$ iff

$$(\forall (\mathbf{x}_i, \mathbf{x}'_i) \in D) \, (\exists j \in [m]) \, \|\mathbf{x}'_i - A_j\mathbf{x}_i\|_\infty \leq \epsilon\|\mathbf{x}_i\|_\infty + \tau. \tag{1}$$

Our approach can be modified to handle $L_2$ or $L_1$ norm instead of $L_\infty$ (see Appendix A).

The switched linear system identification (SLS-ID) problem inputs the number of modes $m \geq 2$, data $D : \{(\mathbf{x}_i, \mathbf{x}'_i)\}_{i=1}^N$, error bound $\tau > 0$, tolerance $\epsilon > 0$ and a magnitude bound $\gamma$. It outputs a set of matrices $A_1, \ldots, A_m$ that fits $D$ with error bound $\tau$ and tolerance $\epsilon$ and satisfies

$$-\gamma E_n \leq A_j \leq \gamma E_n, \ \forall j \in [m], \text{ i.e, each entry of } A_j \text{ lies in } [-\gamma, \gamma], \tag{2}$$

---

[1]Using repeated doubling followed by binary search, we can find the smallest number of modes $m^*$ for which a switched linear system fits the data by running our procedure $O(\log(m^*))$ times.

or outputs INFEASIBLE if no such matrices exist. The SLS-ID problem is known to be NP-hard [22, Theorem 5.1]. We formulate below a mixed-integer linear program (MILP) for the SLS-ID problem.

We choose $mn^2$ decision variables which form the unknown entries of the matrices $A_1, \ldots, A_m$. We also use binary indicator variables $w_{i,1}, \ldots, w_{i,m} \in \{0,1\}$ wherein $i = 1, \ldots, N$ ranges over each data item in $D$. The indicator variable $w_{i,j}$ denotes that the $i^{\text{th}}$ data item in $D$ is fitted by the matrix $A_j$. Each data item must be fitted by one matrix, i.e., $\sum_{j=1}^{m} w_{i,j} = 1$, $\forall\, i \in [N]$. We add constraint (2) to set limits on the magnitude of the entries of the matrices $A_j$. Finally, we add the constraint that each data item is fitted by the corresponding matrix:

$$\|\mathbf{x}'_i - A_j \mathbf{x}_i\|_\infty \leq \epsilon \|\mathbf{x}_i\|_\infty + \tau + (1 - w_{i,j})M\,, \ \forall\, (\mathbf{x}_i, \mathbf{x}'_i) \in D,\ \forall\, j \in [m]\,. \tag{3}$$

Here, $M > 0$ is a constant that is larger than $\gamma \|\mathbf{x}_i\|_1 + \|\mathbf{x}'_i\|_\infty$ for all $(\mathbf{x}_i, \mathbf{x}'_i) \in D$, implying that (3) is trivially satisfied if $w_{i,j} = 0$ (so-called "big-$M$" trick in integer linear programming).

The set of constraints above along with $w_{i,j} \in \{0,1\}$ form the constraints of a MILP. The objective can be set in many ways: for instance, minimize the sum of the absolute value of the entries of each matrix, as a regularization term.

**Lemma 1.** *A set of matrices $A_1, \ldots, A_m$ is feasible for the MILP defined above (for some valuation of the binary variables $w_{i,j}$) iff it fits the data $D$ with error bound $\tau$ and tolerance $\epsilon$.*

The *branch-and-bound* algorithm for solving MILPs will solve $m^N$ linear programs in the worst-case, noting that for each $i \in [N]$ exactly one binary variable in the set $\{w_{i,1}, \ldots, w_{i,m}\}$ can be set to 1. This is exponential in the data size $N\ (= |D|)$. This bound is *seemingly* independent of the dimension $n$ of the underlying state-space. However, if $N < mn$, we are estimating more unknowns than the available data. Therefore, in practice, it generally holds that $N \gg mn$.

## 2.1 Identification with a Tolerance Gap

We now formulate a "relaxed version" of the SLS-ID problem with a "tolerance gap", that will be called the SLS-ID-GAP problem.

*Inputs:* Number of modes $m \geq 2$, data $D : \{(\mathbf{x}_i, \mathbf{x}'_i)\}_{i=1}^{N}$, absolute error bound $\tau > 0$, *two* relative error tolerances $\epsilon_2 > \epsilon_1 \geq 0$ and a magnitude bound $\gamma$. Let $\epsilon_{\mathsf{gap}} = \epsilon_2 - \epsilon_1$. For technical reasons that will be explained later, we assume that $\gamma > \epsilon_{\mathsf{gap}}$.

*Output:* FEASIBLE along with a set of matrices $A_1, \ldots, A_m$ satisfying the bounds constraints (2) that fits $D$ with error bound $\tau$ and tolerance $\epsilon_2$, or, INFEASIBLE if *no* set of $m$ matrices with magnitude bound $\gamma - \epsilon_{\mathsf{gap}}$ can fit the data with *tolerance $\epsilon_1$* and error bound $\tau$.

For any given data set $D$ and tolerance $\epsilon_2 > 0$, we define $\epsilon_{\min}(D, \epsilon_2) \in [0, \infty]$ as the least tolerance $\epsilon$ for which $D$ can be fitted with fixed error bound $\tau$ and magnitude bound $\gamma - (\epsilon_2 - \epsilon)$.[2] Thus, the algorithm for the SLS-ID-GAP problem guarantees that (a) if $\epsilon_{\min}(D, \epsilon_2) \leq \epsilon_1$ then the algorithm will return FEASIBLE with matrices that fit the data with tolerance $\epsilon_2$ and magnitude bound $\gamma$; (b) if $\epsilon_{\min}(D, \epsilon_2) > \epsilon_2$, the algorithm will return INFEASIBLE; (c) if $\epsilon_{\min}(D, \gamma_2) \in (\epsilon_1, \epsilon_2]$, the algorithm may return either answer without any guarantees. The main result of the paper is as follows:

**Theorem 1.** *There exists an algorithm for solving the SLS-ID-GAP problem with complexity $O\big(m^{Cmn^3 |\log(n\gamma/\epsilon_{\mathsf{gap}})|} N \mathsf{poly}(m,n)\big)$, wherein $C$ is a constant factor and $\mathsf{poly}(m,n)$ is polynomial function of $m$ and $n$.*

The significance of this approach is that it is *linear* in the size of the data set $N$, although exponential in the number of modes $m$ and the dimension of the state space $n$.

## 3 Tree-Based Exploration Algorithm

We now present, in stages, our algorithm for finding a set of matrices $A_1, \ldots, A_m$ that fits the data. First, we present the algorithm as a tree-based exploration approach, wherein each node of the tree

---

[2]$\epsilon_{\min}(D, \epsilon_2)$ is well defined; for instance, it suffices to solve the MILP in the previous section with $\epsilon$ as objective function to minimize, and replacing $\gamma$ with $\gamma - (\epsilon_2 - \epsilon)$ in constraint (2). Note that $\epsilon_{\min}(D, \epsilon_2) = \infty$ iff there is $\mathbf{x}_i = 0$ with $\|\mathbf{x}'_i\|_\infty > \tau$ (indeed, this is the only situation in which no set of matrices $A_1, \ldots, A_m$ can fit the data for any tolerance $\epsilon > 0$).

---

**Algorithm 1:** Overall algorithm for switched linear system identification

---
**Data:** $m, D, \tau, \epsilon_1, \epsilon_2, \gamma$ (see Section 2.1).
**Result:** YES with matrices $A_1, \ldots, A_m$ that fit the data with tolerance $\epsilon_2$, or NO.

1   Initialize tree $T$ with a root node (see text for details)
2   **while** *there exist unexplored leaf nodes in $T$* **do**
3      $\nu \leftarrow$ unexplored leaf node in $T$
4      Mark $\nu$ as explored
5      *result* $\leftarrow$ Expand $\nu$ using Algorithm 2
6      **if** *result* $= \langle$FEASIBLE$, (A_1, \ldots, A_m)\rangle$ **then**
7         **return** $\langle$YES$, A_1, \ldots, A_m\rangle$ /* Solution discovered          */

8   **return** NO /* No nodes remain to be explored          */

---

---

**Algorithm 2:** Algorithm to expand a non-terminal leaf node in the tree.

---
**Data:** Leaf node $\nu$ with polyhedra $P_1, \ldots, P_m$, data set $U$ and assignment map $\mu$.
**Result:** New leaf nodes, or matrices $Q_1, \ldots, Q_m$ that fit all the data.

1   **Choose** feasible solutions (matrices) $Q_1, \ldots, Q_m$ s.t. for all $j \in [m]$, $Q_j \in P_j$ (Section 3.2 will specify how to choose $Q_1, \ldots, Q_m$)
2   **Find** $(\mathbf{x}_i, \mathbf{x}_i') \in U$ s.t. for all $j \in [m]$, $\|\mathbf{x}_i' - Q_j\mathbf{x}_i\|_\infty > \epsilon_2 \|\mathbf{x}_i\|_\infty + \tau$
3   **if** *not found* **then**
4      **return** FEASIBLE$, (Q_1, \ldots, Q_m)$    /* $Q_1, \ldots, Q_m$ fit all the data      */
5   **else**
6      $\hat{U} \leftarrow U \setminus \{(\mathbf{x}_i, \mathbf{x}_i')\}$    /* Remove $(\mathbf{x}_i, \mathbf{x}_i')$ from unassigned data      */
7      **for** $j \in [m]$ **do**
         /* Constrain polyhedron $P_j$ s.t. matrix $A_j$ "fits" $(\mathbf{x}_i, \mathbf{x}_i')$      */
8          $\hat{P}_j \leftarrow P_j \cap \{A_j : \|\mathbf{x}_i' - A_j\mathbf{x}_i\|_\infty \leq \epsilon_2\|\mathbf{x}_i\|_\infty + \tau\}$
9          **if** *$\hat{P}_j$ contains a $L_\infty^{\text{ind}}$-norm ball of radius $\epsilon_{\text{gap}}$ (Lemma 3 explains why)* **then**
10             **Create** new child node $\nu_j$ of $\nu$.
11             Associate $\nu_j$ with polyhedra $P_1, \ldots, P_{j-1}, \hat{P}_j, P_{j+1}, \ldots, P_m$
12             $\hat{\mu}_j \leftarrow \mu \cup \{(\mathbf{x}_i, \mathbf{x}_i') \mapsto j\}$    /* Associate $(\mathbf{x}_i, \mathbf{x}_i')$ with mode $j$.      */
13             Associate node $\nu_j$ with set $\hat{U}$ and map $\hat{\mu}_j$

---

represents a set of $m$ convex polyhedra. Subsequently, we will show how a careful choice of the tree exploration strategy will guarantee a bound on the maximum length of each branch of the tree. This, in turn, yields the desired algorithm and its complexity guarantee. The proofs of all the lemmas are outlined: detailed proofs are provided in Appendix B (cf. supplementary material).

The central data structure maintained by our algorithm is a tree $T$. Each node of the tree has the following associated information:

1. Convex polyhedra $P_1, \ldots, P_m$, represented as systems of linear inequalities. Each $P_j \subseteq \mathbb{R}^{n \times n}$ describes a set of possible solutions for matrix $A_j$.

2. A subset $U \subseteq D$ containing data points that have not been "assigned" a mode yet.

3. A assignment map $\mu : (D \setminus U) \to [m]$ mapping each data point in $D \setminus U$ to a mode in $[m]$.

*Root Node:* The root of the tree is initialized by $m$ convex polyhedra $P_1, \ldots, P_m$, wherein $P_j$ encodes the bounds on each entry of the matrix $A_j$: $-\gamma E_n \leq A_j \leq \gamma E_n$. The set $U$ at the root is the full data set $D$ and the associated map $\mu$ is the "empty map" since its domain is empty.

### 3.1 Expanding a Node

Starting from the root node, our algorithm iteratively expands the tree $T$ until no unexplored leaf remains or a suitable set of matrices is found (see Algorithm 1).

*Expansion*: Each expansion step involves choosing an unexplored leaf $\nu$ of the tree and carrying out the steps outlined in Algorithm 2. Namely, expanding $\nu$ begins with choosing feasible solutions $Q_1, \ldots, Q_m$ from polyhedra $P_1, \ldots, P_m$, respectively (line 1). The choice of these solutions will be described in detail in Section 3.2. Next, we scan through all the "unassigned" data $U$ to find a data point $(\mathbf{x}_i, \mathbf{x}_i') \in U$ that cannot be fitted by $Q_1, \ldots, Q_m$ (line 2). If no such data point can be found then we have, in fact, fitted all the data using $Q_1, \ldots, Q_m$. Therefore, we can terminate (line 4). If a data point $(\mathbf{x}_i, \mathbf{x}_i') \in U$ is found in the previous step, then we create $m$ potential child nodes $\nu_1, \ldots, \nu_m$ for the current node. Each potential child $\nu_j$ will have associated polyhedra $P_1, \ldots, P_{j-1}, \hat{P}_j, P_{j+1}, \ldots, P_m$, wherein, for $j' \neq j$, $P_{j'}$ remains the same as in node $\nu$, and $\hat{P}_j$ is defined in order to force the matrix $A_j$ to fit the data point $(\mathbf{x}_i, \mathbf{x}_i')$ that could not be fitted by the previous candidate matrices (line 8). Node $\nu_j$ will have associated data set $\hat{U} : U \setminus \{(\mathbf{x}_i, \mathbf{x}_i')\}$ and mode assignment map $\hat{\mu}_j : \mu \cup \{(\mathbf{x}_i, \mathbf{x}_i') \mapsto j\}$.[3] Finally, the potential child node $\nu_j$ is actually added to the tree only if $\hat{P}_j$ satisfies some lower bound on its size (line 9).

Thus, the process of expanding a node of the tree results either in termination with a feasible solution $Q_1, \ldots, Q_m$ that fits the entire data, or in the possible addition of at most $m$ new leaf nodes to the tree. We will now establish important properties of this algorithm.

**Lemma 2.** *Let $Q_1, \ldots, Q_m$ be the chosen candidates while expanding some node $\nu$. For each child $\nu_j$ of $\nu$ with associated polyhedra $P_1, \ldots, P_{j-1}, \hat{P}_j, P_{j+1}, \ldots, P_m$, we have that $Q_j \notin \hat{P}_j$.*

The proof (see Appendix B) shows that the constraints added to $\widehat{P_j}$ in line 8 are not satisfied by $Q_j$.

Recall that the induced (operator) $L_\infty$ norm of a matrix: $\|A\|_\infty^{\mathsf{ind}} \doteq \max_{\|\mathbf{x}\|_\infty \leq 1} \|A\mathbf{x}\|_\infty$.[4] Given $A \in \mathbb{R}^{n \times n}$ and $\epsilon \geq 0$, let $B_\infty^{\mathsf{ind}}(A, \epsilon)$ denote the ball of radius $\epsilon$ (w.r.t. $\|\cdot\|_\infty^{\mathsf{ind}}$) centered at $A$. We prove that the tree-based exploration algorithm is *complete* for the SLS-ID-GAP problem: if there is a set of matrices $A_1^*, \ldots, A_m^*$ fitting the data with tolerance $\epsilon_1$ and satisfying the reduced magnitude bound $\gamma - \epsilon_{\mathsf{gap}}$, then there is an unexplored leaf whose associated polyhedra contain these matrices.

**Lemma 3.** *At each iteration of the algorithm, there exists an unexplored leaf $\nu$ in the tree with associated polyhedra $P_1, \ldots, P_m$ such that for all $j \in [m]$, $B_\infty^{\mathsf{ind}}(A_j^*, \epsilon_{\mathsf{gap}}) \subseteq P_j$.*

The proof (see Appendix B) is by induction on the number of iterations of the algorithm. At the beginning, we show that the root node satisfies the property of the lemma. Then, we prove that, at each iteration, if a leaf satisfying the property is picked to be expanded, then at least one child of the leaf is created and "inherits" the property. The requirement that $\gamma > \epsilon_{\mathsf{gap}}$ is key for this lemma.

From the definition of the child nodes (line 9), the convex sets associated to each leaf of the tree satisfy a lower bound on their volume.

**Lemma 4.** *At each iteration of the algorithm, and for each leaf $\nu$ in the tree, with associated polyhedra $P_1, \ldots, P_m$, it holds that for all $j \in [m]$, $\mathsf{vol}(P_j) \geq \frac{(2\epsilon_{\mathsf{gap}})^{n^2}}{(n!)^n}$.*

*Proof.* Let $\nu$ be a leaf node as in Lemma 3 and let $j \in [m]$. By Lemma 3, $P_j$ must contain $B_\infty^{\mathsf{ind}}(A_j^*, \epsilon_{\mathsf{gap}})$. The ball $B_\infty^{\mathsf{ind}}(A_j^*, \epsilon_{\mathsf{gap}})$ is the product of $n$ unit $L_1$-norm balls (one for each row) each scaled with a factor $\epsilon_{\mathsf{gap}}$. The volume of a unit $L_1$-norm ball in $n$ dimensions is given by $\frac{2^n}{n!}$. Combining these observations, we have that $\mathsf{vol}(P_j) \geq \left( \frac{(2\epsilon_{\mathsf{gap}})^n}{n!} \right)^n$. $\square$

We have not proven a bound on the size of the tree explored by our algorithm so far. In the next subsection, we describe a way of selecting the candidates $Q_1, \ldots, Q_m$ (line 1) such that the volume of $\hat{P}_j$ will be smaller than some fraction $\alpha < 1$ times the volume of $P_j$. This, combined with the lower bound on the volume of these sets (Lemma 4), will provide a bound on the length of each branch (i.e., the depth) of the tree.

## 3.2 Cutting Plane Argument

We prove an effective bound on the depth of the tree using the so-called *cutting-plane* argument from non-smooth optimization [10]. First, we refine line 1 of our algorithm wherein we choose candidates

---

[3] In other words, the newly assigned data point is assigned to mode $j$ in node $\nu_j$

[4] Note that $\|A\|_\infty^{\mathsf{ind}} = \max(\|A_{1:}\|_1, \ldots, \|A_{n:}\|_1)$ (maximum among the $L_1$ norm of each row) [18].

$Q_j \in P_j$ for each $j \in [m]$. Specifically, we will choose $Q_j$ as the center of the maximum volume inscribed ellipsoid (MVE) of $P_j$. The MVE center of a polyhedron can be computed efficiently by using semi-definite programming (SDP) [6, Proposition 4.9.1]. Now consider the child node $\nu_j$ of $\nu$, to which we associate the polyhedron $\hat{P}_j \subsetneq P_j$. We show that the volume reduces by at least a factor $\alpha \doteq (1 - \frac{1}{n^2}) < 1$.

**Lemma 5.** $\mathsf{vol}(\hat{P}_j) \leq (1 - \frac{1}{n^2})\mathsf{vol}(P_j)$.

The above follows from the classical cutting-plane argument [30] applied on the polyhedron $\hat{P}_j$, which is obtained by cutting $P_j$ with a hyperplane excluding the MVE center (see Appendix B).

Let $V_{\min} = \frac{(2\epsilon_{\mathsf{gap}})^{n^2}}{(n!)^n}$ denote the bound proved in Lemma 4.

**Lemma 6.** *The depth of the tree is $O(mn^4 \log(n\gamma/\epsilon_{\mathsf{gap}}))$.*

The proof (see Appendix B) observes that each time we expand a node in the tree, one polyhedron has its volume reduced by a factor $\alpha$ as stated in Lemma 5. Also, when the overall volume becomes smaller than $V_{\min}$ as in Lemma 4, the node is pruned.

This places a bound on the depth of the tree, as stated in the lemma. In fact, the $n^4$ term is reduced to $n^3$ using the observation that each polyhedron $P_j$ for each node $\nu$ is the Cartesian product of $n$ polyhedra $P_{j,k}$, for $k \in [n]$, involving the variables from the $k^{\text{th}}$ row of the unknown matrix $A_j$. This "fine-grained" analysis is provided in Appendix C.

**Theorem 2.** *The overall size of the tree cannot exceed $m^{O(mn^3 \log(n\gamma/\epsilon_{\mathsf{gap}}))}$ nodes, wherein the complexity of expanding each node is linear in the size $N$ of the data set and involves solving $m$ SDPs each with $n^2$ variables and $O(mn^3)$ constraints.*

## 4  Experimental Evaluation

In this section, we will describe an evaluation of our approach meant to answer two key questions: (a) *Do the theoretical guarantees translate into superior empirical performance when compared to highly optimized MILP solvers?* (Namely, we compare our approach against the MILP solver Gurobi [17] over a set of "microbenchmarks" of varying dimensions, number of modes and data sizes.); (b) *Does the approach yield interesting results on real-life datasets?* (Namely, we illustrate our approach on datasets from handwritten alphabets and mechanical systems with contact forces.)

*Implementation:* We implemented the proposed approach in Python 3.8, using Gurobi [17] to encode and solve linear programs. We assume that the number of modes $m$ (or an upper bound on it) is given. Also, our implementation fixes $\epsilon_1 = 0$ and $\epsilon_2 = \epsilon_{\mathsf{gap}} = \epsilon > 0$. Fixing $\epsilon_1 = 0$ implies that our algorithm either finds matrices that fit the data with tolerance $\epsilon_2$ or concludes that no matrices exist that fit the data with zero *relative* error tolerance. Therefore, throughout this section, we will report the value of $\epsilon$. Our implementation coincides with the description in Section 3 with one important modification: we compute the *Chebyshev center* (center of the largest inscribed *ball*) instead of the center of the maximum volume inscribed ellipsoid (MVE). The Chebyshev center can be computed very efficiently and reliably using Linear Programming [9], and provides a good approximation of the MVE center. Although the theoretical guarantees on the termination of the process using the Chebyshev center are weaker than those with the MVE center (Lemma 5), the use of Chebyshev center in this context is a widely-used heuristic [10, §4.4]. The containment check described in line 9 of Algorithm 2 is implemented by comparing the Chebyshev radius against $\epsilon_{\mathsf{gap}}$.

**Evaluation on Microbenchmarks:** We compare the proposed approach against the MILP approach on a suite of synthetic microbenchmarks. Each such benchmark consists of $m$ randomly chosen $n \times n$ Hurwitz matrices, where $m, n$ are varied systematically. A total of $N$ data points were generated for each experiment from $k = N/T$ trajectories, each of length $T = 10$ time steps starting from a random initial state in $[-1, 1]^n$. An additive noise sampled uniformly at random in the range $[-0.05, 0.05]$ was added to each state. In Figure 1, we illustrate how the two approaches scale in terms of computation time with respect to the number of data points ($N$), number of states ($n$), and number of modes ($m$). All timings are averaged over 10 separate runs to account for the variability in computation times. Further details on microbenchmark generation are provided in Appendix D.

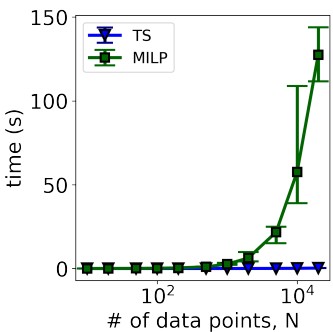 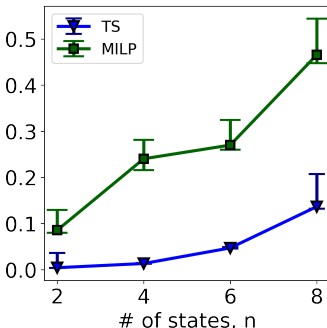 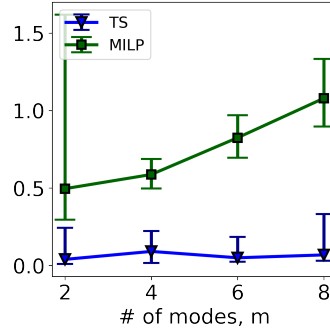

(a) Performance across 10 random microbenchmarks with $n = 4$, $m = 3$, and varying values of $N$.

(b) Performance across 40 random microbenchmarks with $m = 4$, $N = 100$, and varying values of $n$.

(c) Performance across 40 random microbenchmarks with $n = 4$, $N = 200$, and varying values of $m$.

Figure 1: Performance of MILP vs. proposed approach (TS) on a set of microbenchmarks. All timings are reported in seconds on a Linux server running Ubuntu 22.04 OS with 24 cores and 64 GB RAM. Each point in the plot represents the average time taken by the algorithm across 100 experiments (10 runs for each of the 10 microbenchmarks). The error bars represent the minimum and maximum values of time taken across experiments. (a) The proposed approach (TS) scales better than the MILP approach as the number of data points $N$ increases and has smaller variance. Both approaches scale similarly with the dimension (b) and the number of modes (c).

Table 1: Performance of $k$-Linear Regression vs. our approach on a set of microbenchmarks. We use $N$ data points for training both approaches and 50 data points constitute the held-out test dataset. Each row of the table reports average/min/max over 5 runs with $\tau = 0.05$ and $\epsilon = 1$ for a fixed train-test data set. All experiments were carried out on a Linux server running Ubuntu 22.04 OS with 24 cores and 64 GB RAM.

| | | PROPOSED APPROACH | | | | $k$-LINEAR REGRESSION | | | |
| | | Error | | | Time | Error | | | Time |
| | $N$ | avg | max | min | avg (s) | avg | max | min | avg (s) |
|---|---|---|---|---|---|---|---|---|---|
| Benchmark #1 | 20 | 0.17 | 1.16 | 0.01 | 0.04 | 0.34 | 3.21 | 0.01 | 0.03 |
| ($n = 2, m = 4$) | 100 | 0.05 | 0.33 | 0.00 | 8.54 | 0.23 | 2.13 | 0.01 | 0.02 |
| | 500 | 0.05 | 0.37 | 0.00 | 15.61 | 0.19 | 2.15 | 0.00 | 0.02 |
| Benchmark #2 | 20 | 0.14 | 1.10 | 0.01 | 2.62 | 0.29 | 2.13 | 0.02 | 0.08 |
| ($n = 4, m = 2$) | 100 | 0.07 | 0.46 | 0.01 | 1.37 | 0.21 | 2.31 | 0.02 | 0.02 |
| | 500 | 0.06 | 0.26 | 0.02 | 1.56 | 0.27 | 2.47 | 0.02 | 0.03 |
| Benchmark #3 | 20 | 0.56 | 2.72 | 0.03 | 1.70 | 0.95 | 5.58 | 0.02 | 0.16 |
| ($n = 4, m = 4$) | 100 | 0.23 | 1.65 | 0.03 | 506.29 | 0.48 | 3.74 | 0.01 | 0.41 |
| | 500 | 0.18 | 1.24 | 0.02 | 1184.16 | 0.41 | 2.46 | 0.02 | 0.11 |

The comparison clearly shows that the MILP solver's computation time increases rapidly with the number of data points $N$. In contrast, the running time of our approach depends linearly on $N$. Both approaches scale similarly in terms of number of modes ($m$) and dimension ($n$).

We compare the accuracy of the proposed approach against $k$-*Linear Regression* ($k$LR) [21] in terms of one-step prediction errors: $\min_{j=1}^{m} ||\mathbf{x}_i' - A_j \mathbf{x}_i||_\infty$, measured over a held out test dataset. Table 1 compares the prediction errors for a set of 3 microbenchmarks. The proposed approach has smaller error bounds compared to $k$LR but takes more time, as expected.

**Handwriting Recognition:** We now evaluate our approach on a dataset from human handwriting on a tablet: our goal is to identify various modes with dynamics that describe how letters are traced out on the tablet. We generate our own handwriting dataset by having an author trace out the letters "a", "b", "c" and "d" using their fingers on the mouse pad of their laptop. We collect the $(x, y)$ locations of the handwritten letters over time. We generate our final dataset by interpolation in order to ensure that the samples are roughly equidistant from each other. This can be done in an online fashion, if need be. It ensures that our model here is not capturing artifacts of the pressure sensor in

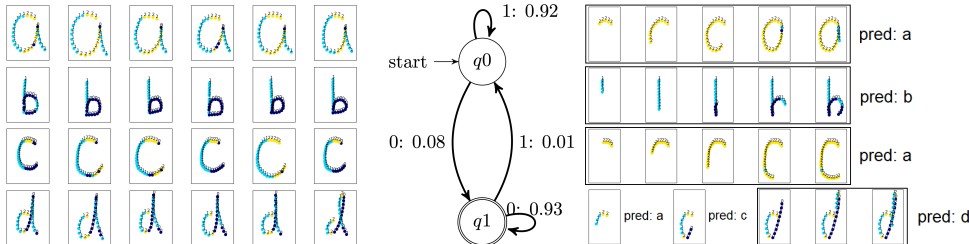

Figure 2: **Left:** Handwriting samples with identified modes shown in different colors, **Middle:** JMM model for letter b and **Right:** predicting letters from partial observations using the JMM.

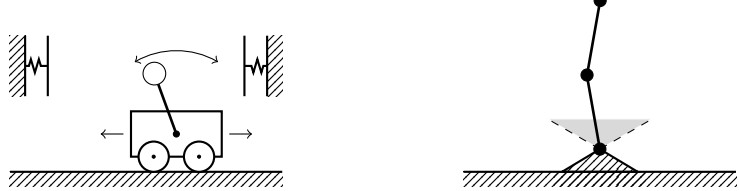

Figure 3: **Left:** Cartpole with soft walls. **Right:** Acrobot with soft joint limits.

the mouse pad or how the person tracing the alphabet may speed-up or slow-down at various points in time. We also scale the $(x, y)$ values so that the letters fit within a square of side length 1. Further details are provided in Appendix F. We collect 10 samples for each letter (Figure 2-left).

We apply our approach to learn $m = 3$ matrices that fit the data with $\epsilon = 0.05$ and $\tau = 0.1$. For each sample, we obtain a corresponding switching sequence from this process. We apply a well-known grammatical inference approach to learn a Jump Markov Model (JMM), wherein the various dynamical modes are represented on the edges of the automaton [11, 34]. The estimated mode for each data point of each letter is represented in Figure 2-left. Each color represents a different mode. We note that the modes naturally corresponds to different parts of the letters.

Figure 2-middle shows an example of JMM for the letter "b". Each transition in the JMM represents a mode and a probability of taking the edge. At state $q_0$, mode 1 is observed for $92\%$ of the time, and at $q_1$, mode 0 is observed for $93\%$ of the time. We then try to predict a class for an alphabet as it is being written, based on the estimated JMMs (see Figure 2-right). Each column represents $20\%$, $40\%$, $60\%$, $80\%$, $95\%$ portion of a letter. We use the JMMs and Bayesian inference to compute the probability of a letter given a partial sequence of states. The letters "a", "b" and "d" are correctly predicted but the letter "c" is classified as an "a": a visual inspection of Figure 2-right explains why this is the case.

In conclusion, we show that our switched linear system identification algorithm works on the noisy handwriting data and can be used to identify a JMM from sequences of identified modes which in turn can be useful for data classification and prediction.

**Acrobot:**  We evaluate our approach on an acrobot benchmark with soft joint limits [3]. We sampled $N = 30$ trajectories with time step $dt = 0.03$ seconds for $T = 3$ seconds. We then identified the dynamics at each mode using the proposed approach (with $m = 3$, $\tau = 0.01$, and $\epsilon = 0.01$). The average one-step prediction error[5] of the proposed approach on a held-out test dataset consisting of 5 trajectories is 0.005 (min: 0.0, max: 0.18). The average training time is 7.56 seconds. In comparison, the average one-step prediction error of $k$-Linear Regression($k$LR) [21] on the same dataset is 1.80 (min: 0.0, max: 8.68). The average training time of $k$LR is 0.18 seconds. Figure 4 shows the predictions of the proposed approach on a sample test trajectory. Despite the underlying system having an infinite number of modes (as the time-sampled system of a continuous-time hybrid linear system), our system identification technique is able to identify three main linear modes that explain most of the data both in the training and test datasets.

---

[5]One-step prediction error: $\min_{j=1}^{m} ||\mathbf{x}_i' - A_j \mathbf{x}_i||_\infty$

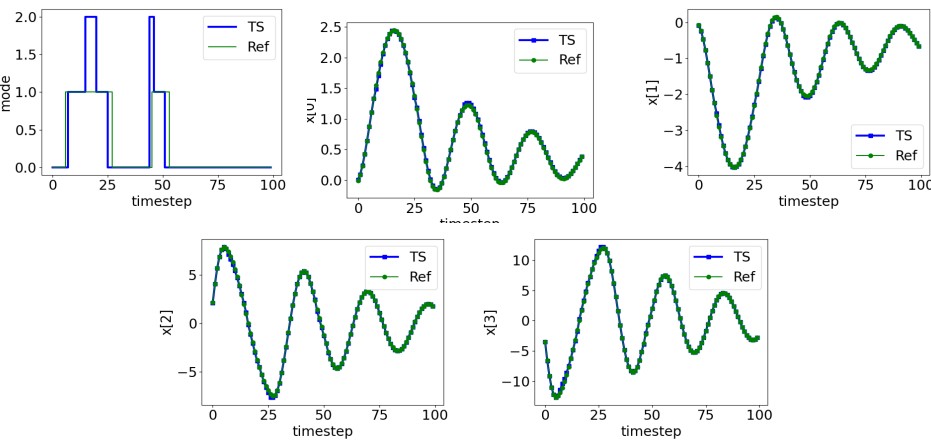

Figure 4: Data from the Acrobot system plotted against those of the identified model.

**Cartpole:** We evaluate our approach on a cartpole system with soft walls [3]. We sampled $N = 30$ trajectories with time step $dt = 0.05$ seconds for $T = 5$ seconds. We then identified the dynamics at each mode using the proposed approach (with $m = 3$, $\tau = 0.1$, and $\epsilon = 0.1$). The average one-step prediction error of the proposed approach on a held-out test dataset consisting of 5 trajectories is 0.016 (min: 0.00, max: 0.15). The average training time is 9.35s. The average prediction error of $k$-Linear Regression($k$LR) [21] is 0.04 (min: 0.0, max: 0.15) and the average time taken by $k$LR is 0.16 seconds. Additional plots are provided in Appendix E. Similarly to the acrobot example, the learned system is able to explain most of the data (both in the training and test datasets) with a few (three) linear modes, despite the fact that the actual system has an infinite number of modes (as the time-sampled system of a continuous-time hybrid linear system).

## 5 Conclusions: Limitations, Future Work and Broader Impacts

In summary, we have presented an approach that improves the time complexity of switched linear system identification using a novel problem formulation with a "gap" and using ideas from the ellipsoidal method in combinatorial optimization. The resulting complexity is linear in the number of data points. Limitations include (a) Our approach uses full state information and extends to output observations using SARX models. However, the number of modes in the SARX model can be exponentially larger when compared to the state-space model. (b) Our approach finds the matrices fitting dynamics but does not infer the "generator" for the mode switches, although we present an example using Jump Markov Models. However, the problem of learning the matrices for various modes in conjunction with the "generator" may not always lend itself to a clean separation wherein the modes are first inferred using our approach and a generative model such as a Markov chain is then inferred. For instance, our approach may find matrices that fit the data well but lead to poorly fitting system models. We will study the use of backtracking in our approach to generate multiple "dissimilar" solutions; (c) Finally, our implementation performs the tree exploration without leveraging other information about the system or using heuristics that have proved powerful in branch-and-bound algorithms. Nevertheless, it provides a proof of concept that it can be competitive against highly-optimized MILP solvers. We will expand on these ideas in our future work.

In terms of broader impacts, model identification is a very important machine learning problem that has very important and beneficial applications in areas such as controls, autonomous systems and medical applications. Better modeling of physiological processes can lead to improved closed-loop medical devices. Improved ability to predict human movements can lead to safer human-robot interactions. However, our approach can be used to improve surveillance systems that could be themselves be used in ways that are detrimental to the freedom of individuals and societies.

**Acknowledgments:** We thank the anonymous reviewers for their detailed comments and suggestions. This research was funded in part by the Belgian-American Education Foundation (BAEF) and the US National Science Foundatton (NSF) under award numbers 1836900 and 1932189.

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
