# A    Problem Formulation using $L_1$ and $L_2$ norms

Our approach essentially amounts to computing convex sets $P_j \subseteq \mathbb{R}^{n \times n}$ containing the feasible values for the matrices $A_j$. If the $L_\infty$ or $L_1$ norm is used for the constraint (1) on $A_j$, then the sets $P_j$ are polyhedra. In fact, if the $L_\infty$ norm is used, the set $P_j$ can even be described as the Cartesian product of $n$ polyhedra in $\mathbb{R}^n$ (one for each row of $A_j$). If the $L_2$ norm is used in (1), then the convex sets can be described using second-order cones and linear constraints.

In any case, that is, for any vector norm $\|\cdot\|$, the convex set $P_j$ defined by the constraints (1) with the appropriate norm satisfies that $B^{\mathrm{ind}}(A_j^*, \epsilon_{\mathsf{gap}}) \subseteq P_j$, where $\epsilon_{\mathsf{gap}} = \epsilon - \epsilon_1$ and $A_j^*$ explains the data with error bound $\tau$ and tolerance $\epsilon_1$. Here, $B^{\mathrm{ind}}(A_j^*, \epsilon_{\mathsf{gap}})$ is the ball in $\mathbb{R}^{n \times n}$ centered at $A_j^*$ with radius $\epsilon_{\mathsf{gap}}$ w.r.t. the matrix norm $\|\cdot\|^{\mathrm{ind}}$ induced by $\|\cdot\|$ (the proof is identical to the one of Lemma 3). Lemma 4 then applies *mutatis mutandis* using the volume of $B^{\mathrm{ind}}(0, \epsilon_{\mathsf{gap}})$.

Finally, regarding the computation of the MVE centers of the sets $P_j$ (which is a key step of Algorithm 2, as it is used to compute the candidate matrices $A_j$ in line 1): finding the MVE center of convex sets described by linear and second-order cone constraints can be cast as a semidefinite optimization problem [9, § 8.2.4], so that it can be solved efficiently.

# B    Missing Proofs From Section 3

*Proof of Lemma 2.* Let $U$ be the data set associated to $\nu$. Since we did not terminate, there is $(\mathbf{x}_i, \mathbf{x}_i') \in U$ such that $\|\mathbf{x}_i' - Q_j \mathbf{x}_i\|_\infty > \epsilon_2 \|\mathbf{x}_i\|_\infty + \tau$. However, $\hat{P}_j$ is obtained by adding the constraint $\|\mathbf{x}_i' - A_j \mathbf{x}_i\|_\infty \leq \epsilon_2 \|\mathbf{x}_i\|_\infty + \tau$ to the existing constraints in $P_j$. Therefore, $Q_j \notin \hat{P}_j$ since it violates the new constraint. $\qquad\square$

*Proof of Lemma 3.* First, we prove that the property holds for the root node. Therefore, let $A_j \in B_\infty^{\mathrm{ind}}(A_j^*, \epsilon_{\mathsf{gap}})$. By the formula for the induced norm, it holds that for any matrix $M$, if $\|M\|_\infty^{\mathrm{ind}} \leq \epsilon$, then for all $k_1, k_2 \in [n]$, $|M_{k1,k2}| \leq \|M_{k_1:}\|_1 \leq \epsilon$, so that $-\epsilon E_n \leq M \leq \epsilon E_n$. Hence, letting $M \doteq A_j - A_j^*$ and $\epsilon \doteq \epsilon_{\mathsf{gap}}$, we get that $-\epsilon_{\mathsf{gap}} E_n \leq A_j - A_j^* \leq \epsilon_{\mathsf{gap}} E_n$. Thus, $-\gamma E_n \leq A_j \leq \gamma E_n$, so that $A_j \in P_j$ at the root node.

Suppose that at the beginning of the $k^{\mathrm{th}}$ iteration, the unexplored leaf $\nu$, with associated polyhedra $P_1, \ldots, P_m$, has the property that $B_\infty^{\mathrm{ind}}(A_j^*, \epsilon_{\mathsf{gap}}) \subseteq P_j$ for all $j \in [m]$. We wish to prove the property for some unexplored leaf after the iteration. This is trivial if the leaf $\nu$ is not expanded in that iteration. Suppose the leaf $\nu$ is expanded. Let $(\mathbf{x}_i, \mathbf{x}_i')$ be the data point that cannot be explained by the candidates that were chosen, and let $j \in [m]$ be such that $A_j^*$ explains the data point $(\mathbf{x}_i, \mathbf{x}_i')$ with tolerance $\epsilon_1$. We show that $B_\infty^{\mathrm{ind}}(A_j^*, \epsilon_{\mathsf{gap}}) \subseteq \hat{P}_j$ where $\hat{P}_j$ is defined as in line 8. Indeed, let $A_j \in B_\infty^{\mathrm{ind}}(A_j^*, \epsilon_{\mathsf{gap}})$. It holds that $\|\mathbf{x}_i' - A_j \mathbf{x}_i\|_\infty \leq \|\mathbf{x}_i' - A_j^* \mathbf{x}_i\|_\infty + \|(A_j - A_j^*)\mathbf{x}_i\|_\infty \leq \|\mathbf{x}_i' - A_j^* \mathbf{x}_i\|_\infty + \epsilon_{\mathsf{gap}} \|\mathbf{x}_i\|_\infty \leq \epsilon_1 \|\mathbf{x}_i\|_\infty + \tau + \epsilon_{\mathsf{gap}} \|\mathbf{x}_i\|_\infty \leq \epsilon_2 \|\mathbf{x}_i\|_\infty + \tau$, where the second inequality comes from the definition of the induced norm and the assumption on $A_j$, and the third inequality comes from the assumption on $A_j^*$ explaining $(\mathbf{x}_i, \mathbf{x}_i')$ with tolerance $\epsilon_1$. Thus, $B_\infty^{\mathrm{ind}}(A_j^*, \epsilon_{\mathsf{gap}}) \subseteq \hat{P}_j$. Since, $B_\infty^{\mathrm{ind}}(A_j^*, \epsilon_{\mathsf{gap}}) \subseteq P_j$, this implies that $B_\infty^{\mathrm{ind}}(A_j^*, \epsilon_{\mathsf{gap}}) \subseteq \hat{P}_j$, concluding the proof. $\qquad\square$

*Proof of Lemma 5.* From Lemma 2, we note that $Q_j \notin \hat{P}_j$. In other words, $\hat{P}_j \subsetneq P_j$ excludes the MVE center of $P_j$. Following [30] (or [10, § 4.3] for a more recent reference), we have $\mathrm{vol}(\hat{P}_j) \leq (1 - \frac{1}{d})\mathrm{vol}(P_j)$, where $d$ is the dimension of $P_j$. Here, $d = n^2$, concluding the proof. $\qquad\square$

*Proof of Lemma 6.* Consider any path from the root to a leaf whose length is $mK$ for some integer $K > 0$. We note that for each node $\nu$ and any of its children $\nu_j$, the polyhedron $\hat{P}_j$ satisfies the inequality $\mathrm{vol}(\hat{P}_j) \leq \alpha \mathrm{vol}(P_j)$, where $\alpha = 1 - \frac{1}{n^2}$ (Lemma 5). Let us say that the index $j \in [m]$ is *refined* by such an edge. By the pigeon-hole principle, for a path of length $mK$, there exists at least one index $j$ that is refined $K$ or more times along the path. Therefore, we have that: $\mathrm{vol}(P_j^{(K)}) \leq \alpha^K \mathrm{vol}(P_j^{(0)})$, where $P_j^{(0)}$ is the $j^{\mathrm{th}}$ polyhedron at the root and $P_j^{(K)}$ is the $j^{\mathrm{th}}$ polyhedron at the leaf.

We know that $\text{vol}(P_j^{(0)}) = (2\gamma)^{n^2}$. Thus, there exists $K_{\min}$ such that for any $K \geq K_{\min}$, $\text{vol}(P_j^{(K)}) < V_{\min}$ and thus the branch will end up being "pruned" by our algorithm (line 9). It holds that

$$K_{\min} \leq \frac{\log((2\gamma)^{n^2}) - \log(V_{\min})}{-\log(\alpha)} \leq \frac{\log((2\gamma)^{n^2}) - \log((2\epsilon_{\text{gap}})^{n^2}) + \log(n^{n^2})}{-\log(\alpha)} \leq n^4 \log\left(\frac{n\gamma}{\epsilon_{\text{gap}}}\right),$$

where the last inequality follows from $\log(1 - \frac{1}{n^2}) \leq -\frac{1}{n^2}$. Therefore, the depth is upper bounded by $mK_{\min} = mn^4 \log(n\gamma/\epsilon_{\text{gap}})$. □

## C  Fine-Grained Complexity Analysis of Tree Search

As we noticed in Appendix A, for the $L_\infty$ norm, each set $P_j$ can be described as the Cartesian product of $n$ polyhedra in $\mathbb{R}^{1 \times n}$ (one for each row of the matrix). The MVE center of a Cartesian product of convex sets is the vector containing the MVE center of each convex set. Therefore, the volume reduction guarantee in Lemma 5 can be refined as: $\text{vol}(\hat{P}_j) \leq (1 - \frac{1}{n})\text{vol}(P_j)$ (see Lemma 8 below). By applying the same argument as in the proof of Lemma 6, we then get the bound $O(mn^3 \log(n\gamma/\epsilon_{\text{gap}}))$ on the depth of the tree.

We will now present this in more details. Let $\nu$ be any node of the tree and $P_1, \ldots, P_m$ be the associated polyhedra.

**Lemma 7.** *Each $P_j$ can be written as a Cartesian product $P_j = P_{j,1} \times \cdots \times P_{j,n}$ wherein each polyhedron $P_{j,i} \subseteq \mathbb{R}^{1 \times n}$ involves just those decision variables of the matrix $A_j$ associated with its $i^{th}$ row.*

*Proof.* Proof is by induction. To begin with, we note that this is true for the root node of the tree. For the induction step, assume that the property is satisfied at some node $\nu$ of the tree and consider any of its child nodes $\nu_j$. We note that, for any $(\mathbf{x}_i, \mathbf{x}_i')$, the constraint $\|\mathbf{x}_i' - A_j\mathbf{x}_i\|_\infty \leq \epsilon_2\|\mathbf{x}_i\|_\infty + \tau$ is of the form $\|\mathbf{z}\|_\infty \leq a$ for a vector $\mathbf{z}$ and scalar $a$. This can be decomposed into constraints $-a \leq \mathbf{z}_i \leq a$ for each row of $\mathbf{z}$. Hence, the constraint $\|\mathbf{x}_i' - A_j\mathbf{x}_i\|_\infty \leq \epsilon_2\|\mathbf{x}_i\|_\infty + \tau$ can be decomposed into a conjunction of $n$ constraints, each involving a different row of $A_j$. From the induction hypothesis and the definition of $\hat{P}_j$ (line 8), it follows that $\hat{P}_j$ can be written as a Cartesian product of $n$ polyhedra, each involving a different row of $A_j$, concluding the proof. □

We now state a refined version of Lemma 5.

**Lemma 8.** $\text{vol}(\hat{P}_j) \leq (1 - \frac{1}{n})\text{vol}(P_j)$.

*Proof.* Let $\hat{P}_{j,i}$ be the polyhedron associated with the $i^{th}$ row of matrix $A_j$ in the $j^{th}$ child node of some node $\nu$ such that the $Q_{j,i} \notin \hat{P}_{j,i}$, wherein $Q_{j,i}$ denotes the $i^{th}$ row of candidate matrix $Q_j$ explored during the expansion of node $\nu$ by Algorithm 2. Since $Q_{j,i}$ is the MVE center of $P_{j,i}$, it holds, by the cutting-plane argument (cf. proof of Lemma 5), that $\text{vol}(\hat{P}_{j,i}) \leq (1 - \frac{1}{n})\text{vol}(P_{j,i})$. Now, since $\text{vol}(\hat{P}_j) = \prod_{i=1}^{n} \text{vol}(\hat{P}_{j,i})$ and $\text{vol}(P_j) = \prod_{i=1}^{n} \text{vol}(P_{j,i})$, we get the desired result. □

**Lemma 9.** *The depth of the tree is $O(mn^3 \log(n\gamma/\epsilon_{\text{gap}}))$.*

*Proof.* The proof is very similar to that of Lemma 6. The only thing that to be changed is the value of $\alpha = 1 - \frac{1}{n}$ (instead of $1 - \frac{1}{n^2}$). We can then use the bound: $\log(1 - \frac{1}{n}) \leq -\frac{1}{n}$, to get the desired result. □

## D  Details on the Microbenchmarks

For Figure 1, we generate 90 microbenchmarks with varying values of $n$ and $m$. We fixed the dynamics at each mode of the microbenchmark by sampling a random $n \times n$ Hurwitz matrix. The Hurwitz matrices were generated by first generating random diagonal and invertible matrices of appropriate dimensions and then applying a similarity transformation on them. We then generated

trajectories from the microbenchmark by starting at some initial state in $[-1, 1]^n$ and simulating the forward in time for T=10 time steps by randomly picking the mode at each time step. We added a uniform noise with amplitude $\in [-0.05, 0.05]$ to all the trajectories. Figure 5 shows one such microbenchmark with $n = 4$ and $m = 3$ and some sample trajectories from the microbenchmark.

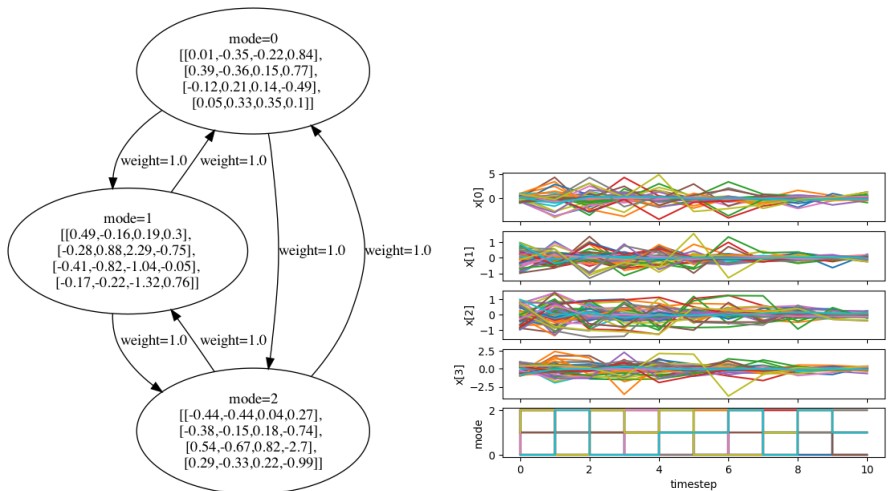

Figure 5: Microbenchmark (**Left**) with $n = 4, m = 3$ and sample trajectories (**Right**).

## E    Details on the Acrobot and Cartpole Benchmarks

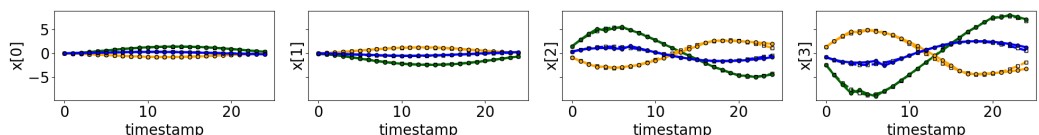

Figure 6: 3 trajectories of the Acrobat benchmark: The dashed lines with square markers show the reference trajectories. The solid lines with triangle markers show the trajectories predicted using the dynamics identified by the proposed approach.

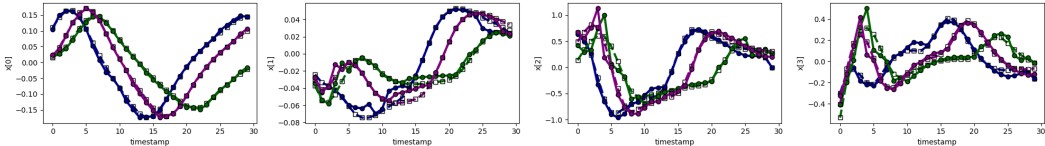

Figure 7: 3 trajectories of the Cartpole benchmark: The dashed lines with square markers show the reference trajectories. The solid lines with triangle markers show the trajectories predicted using the dynamics identified by the proposed approach.

## F    Details on the Handwriting Recognition

We generated a human handwriting dataset where each letter is drawn on a canvas of size $300 \times 300$ pixels. Therefore, we asked the users to write letters $a, b, c, d$ and collected the locations $(x, y)$ of the handwritten letters. We made sure that each handwritten letter had enough raw points ($T_{\mathrm{raw}} > 100$). Subsequently, we interpolated the raw data points so that we could extract $T = 30$ data points that are roughly equidistant from each other. The complete interpolation and extraction process in explained below; see also Fig. 8 for an illustration.

Given the raw data points $P : \{p_1, ..., p_{T_{\mathrm{raw}}}\}$, with $p_i : (x_i, y_i) \in \mathbb{R}^2$,

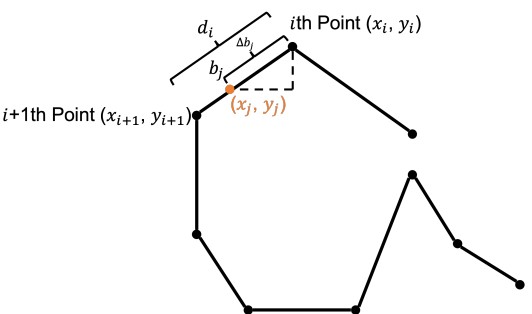

Figure 8: The schematic of computing the equidistant points.

1. Measure the total distance $D = \sum_i d_i$ where $d_i = \sqrt{(x_{i+1} - x_i)^2 + (y_{i+1} - y_i)^2}$

2. For each $j \in \{0, \ldots, T-1\}$, let $d'_j = \frac{D}{T-1}j,$ .

3. Compute the $j^{\text{th}}$ data point $(\hat{x}_j, \hat{y}_j)$, as the point on the line that is at a distance $d'_j$ from $p_1$. Therefore:

    (a) Find $i \in \{1, \ldots, T_{\text{raw}} - 1\}$ such that $(\hat{x}_j, \hat{y}_j)$ is located between $p_i$ and $p_{i+1}$, i.e., such that the following condition holds true,

    $$\sum_{i=0}^{i} d_i \leq d'_j < \sum_{i=0}^{i+1} d_i.$$

    (b) Define $(\hat{x}_j, \hat{y}_j) : (1-t)p_i + tp_{i+1}$, where $t = (d'_j - \sum_{i=0}^{i} d_i)/d_i$. Since $(\hat{x}_j, \hat{y}_j) = p_i + t(p_{i+1} - p_i)$, this satisfies that requirement that $(\hat{x}_j, \hat{y}_j)$ is at a distance $d'_j$ from $p_1$.

We collected 10 samples for each letter and each letter's final data size was set to $T = 30$. As a result, the total number of data points is $N = kT = 10 \cdot 30 = 300$.