# OpenReview forum: "An Algorithm for Learning Switched Linear Dynamics from Data"
_NeurIPS.cc/2022/Conference — NeurIPS 2022 Accept_

### Official Review · Reviewer_h6Sd · 2022-07-10

**Rating:** 7
**Confidence:** 3
**Soundness:** 3 good
**Presentation:** 3 good
**Contribution:** 3 good

**Summary:**

Summary: This paper studies the identification of switched linear dynamics where the underlying mode is hidden (but the total number of modes is known). The contribution is an approach whose computation complexity scales linearly in the number of samples, as opposed to the exponential scaling in the MILP approach. However, the complexity of the proposed approach still scales exponentially with the number of modes and the state dimension. The approach is a tree search algorithm, and the core innovation is a pruning-based technique on the volume. This limits the depth of the tree. The computation complexity of the approach is provided.


**Questions:**

Line 4 of algorithm 1: 1: how are the $Q_1,\ldots,Q_m$ chosen. Arbitrarily from $P_j$?

Line 6 of algorithm 1, I assume $(x_i,x_i’)$ is from line 2. However, what if Line 2 returns multiple pairs of $(x_i,x_i’)$?

Line 232: should it be the mode $j = \mu^*(x_i,x_i’)\in [m]$?

Line 224: where does the $2^{mN}$ compute from?  Based on my read of the algorithm, at each layer, each node splits into m nodes, and the maximum depth is N. So shouldn’t the complexity be $m^N$?

Table 1, benchmark 2, tau = 0.00, for the TS run time, why the run time for N=10000 is smaller than N=1000.

Line 236: a “stronger” assumption is introduced here - does the main result Theorem 1 relies on this assumption? If so, this stronger assumption should be introduced in or before Theorem 1.

In both Lemma 4 and Lemma 5: there is a notion of “iteration”, which I think means each round of expansions of the current leaf nodes. Also, in Line 257, there is a redefinition of a step of the algorithm. Going back to my earlier comment on pseudo-code, I think the paper needs a formal presentation of the full algorithm so that the “iteration” is formally defined and the details of the algorithm are finalized.


**Limitations:**

The authors have addressed the limitations and potential negative societal impact of their work

**Strengths And Weaknesses:**

Weakness:

Still, exponential dependence on the number of nodes and state dimension is needed. The author needs to provide a discussion on why the exponential dependence is difficult to avoid. For example, I don’t really see why the exponential dependence on $n$ is needed. Based on the analysis, the exponential dependence comes from Lemma 6, but is there a way around it?  Also, the MILP algorithm only scales exponentially with $m,N$, not $n$, right? If so, this means the MILP algorithm could scale to high dimensions, while the proposed algorithm cannot. This should be acknowledged and discussed.

Comparison: the majority of the paper focuses on comparison with the MILP approach, which is fine. However, in the related work session, a few other approaches are mentioned, e.g. the Ozay et al one and the clustering approach. How does the proposed approach compare with these related work, both theoretically and empirically?

Presentation: the only pseudo-code is Algorithm 1, which is a subroutine on how to expand a node in the search tree. I suggest adding the pseudocode for the entire algorithm.

---

> ### Author Response · Authors · 2022-08-02
> **Reviewer h6Sd**
>
> Thank you for your insightful comments: we appreciate your feedback. Our response to your comments are inlined below.
>
> >  Based on the analysis, the exponential dependence comes from Lemma 6, but is there a way around it? The MILP algorithm only scales exponentially with m,N, not n, right? If so, this means the MILP algorithm could scale to high dimensions, while the proposed algorithm cannot. This should be acknowledged and discussed.
>
> Your observation is correct but we can see no way around it unless we assume something about the switching sequence (such as bounded number of switches or dwell-time constraints). The MILP algorithm scales exponentially with $m$, $N$ but not $n$. However, note that if the number of data points is $N < m \times n$, the problem becomes under-constrained because you have $m \times n^2$ parameters to estimate but $n \times N$ numbers in the data. Therefore, in most "meaningful" cases with error/noise, we need to have $N \gg m \times n$, or else the problem does not have enough data when compared to the number of parameters.
>
> In summary, we would expect the number of time steps $N$ to dominate $m \times n$, in practice.
>
> > However, in the related work session, a few other approaches are mentioned, e.g. the Ozay et al one and the clustering approach. How does the proposed approach compare with these related work, both theoretically and empirically?
>
> The clustering-based approach has no theoretical comparison and many of the works cited rely on the switching sequences having special properties. For instance, one of Ozay et al’s cited works ([23]) relies on the number of mode switches being bounded to achieve a polynomial-time algorithm. Another relies on a moment-based approximation ([24]) for solving the MILP approximately. Clustering approaches to this problem are very similar in spirit to k-means clustering: they do not derive any optimality guarantees, and often get stuck in a local saddle point.
>
> In summary,  the techniques mentioned in related work are not "apples-to-apples" comparable with our method. For all these reasons, we did not perform an empirical comparison with our approach.
>
> > The only pseudo-code is Algorithm 1
>
> Thank you for the suggestion: we will add the overall algorithm pseudo-code as well in our final version. If accepted, we would get an extra page that will be sufficient to have this addition.
>
> > How are $Q_1, \ldots, Q_m$ chosen? Arbitrarily from $P_j$?
>
> At that point in the presentation, we did not want to specify this detail. A little later in the same section (top paragraph of page 7), we clarify that $Q_j$ is chosen as the Max-Volume Inscribed Ellipsoid (MVE) center of the polyhedron $P_j$.
>
> > However, what if Line 2 returns multiple pairs of $(x_i, x_i’)$?
>
> We assume that Line 2 returns one such pair and if there are more than one, it arbitrarily chooses one. For the correctness of the algorithm, it does not matter which one is chosen.
>
> > Line 232: should it be the mode $j = …$?
>
> That is correct. We will fix the confusion between "$:$" and "$=$".
>
> > Line 224: where does the $2^{mN}$ compute from? Based on my read of the algorithm, at each layer, each node splits into m nodes, and the maximum depth is N. So shouldn’t the complexity be $m^N$?
>
> This is correct. $2^{mN}$ is an upper bound obtained from the fact that we have $mN$ binary variables in the MILP solver and in the worst case the complexity of binary MILP requires solving $2^{mN}$ linear programs. However, if we just implemented a search method, the complexity will be $m^N$. We will clarify and fix this in the revised version.
>
> > Table 1, benchmark 2, tau = 0.00, for the TS run time, why the run time for N=10000 is smaller than N=1000.
>
> We double checked those timings.  Even though N=10,000 represents more data, the tree search algorithm in this instance was able to rapidly identify the matrices for the modes by exploring fewer nodes in the tree (involving a small subset of the data)  in order to explain away all the data. However, for N=1,000 the algorithm on this particular instance needed to explore more of the tree before it could find matrices that explained away all the data.
>
> > Line 236: a "stronger" assumption is introduced here - does the main result Theorem 1 relies on this assumption? If so, this stronger assumption should be introduced in or before Theorem 1.
>
> Yes, this constraint is needed for theorem 1. We tried to do this to keep the presentation less confusing but you are right: we will emphasize this in theorem 1.
>
> > In both Lemma 4 and Lemma 5: there is a notion of "iteration", which I think means each round of expansions of the current leaf nodes. Also, in Line 257, there is a redefinition of a step of the algorithm. Going back to my earlier comment on pseudo-code, I think the paper needs a formal presentation of the full algorithm so that the "iteration" is formally defined and the details of the algorithm are finalized.
>
> We will be adding that to our paper. Thank you.

---

### Official Review · Reviewer_FbXU · 2022-07-10

**Rating:** 3
**Confidence:** 4
**Soundness:** 2 fair
**Presentation:** 1 poor
**Contribution:** 2 fair

**Summary:**

This work deals with the problem of learning a switched linear system model from state-trajectory data, where the underlying mode switches are not observed. The goal is to learn state transition matrices and an assignment of each transition in a given data set to the corresponding mode, i.e., linear transition function, such that a certain error criterion is fulfilled.
Since the naive problem setting based on Mixed-Integer-Linear-Programming is computationally infeasible, the authors propose a relaxation of the problem. For the relaxed setting, a concrete algorithm is presented, its correctness is proven and its time complexity is investigated. Finally, the algorithm is applied to synthetic task, a handwriting prediction task and a traffic data set.

**Questions:**

## Questions
* How exactly could the SARX model be used to deal with output measurements? Does the algorithm and its analysis need adaption to this case?
* Page 5, line 203: "then we create $m$ child nodes" This seems incorrect, since in Algorithm 1 a child node $\nu_j$ is only created if $\hat{P}_j$ is feasible
* Proof of Lemma 4, lines 531-532: Does such a leave node (with $\mu=\mu^\ast$) always exists?
* Proof of Lemma 5: According to the proof, the lower bound should be $\epsilon_\text{gap}^n2^{n^2}/(n!)^n$ How is the lower bound stated in the lemma then derived?
* Experiments:
  - Why the choice of $\epsilon_1=0$? Does it make the problem easier?
  - the MVE is replaced by Chebychev centers. What exactly is lost when using Chebychev centers, i.e., which guarantees cannot be given anymore?
  - What is the MILP approach? No details are given
* Page 7, line 303: Why $N=kT$? If the trajectories have $T$ time steps, then there are only $T-1$ transitions (and each data point correponds to a transition)

## Additional detailed remarks
* p1, l26: Why "explain away"? This is somewhat of a technical term in e.g. Bayesian networks
* p1, l28: Reference for NP-hard would be good here
* p1, l30: Reference for exponential complexity would be good here
* p2, l37: Unclear, what is $\epsilon_2$?
* p2, l39: The rest of the paragraph is very unclear. E.g., it is unclear what is meant by "tree" here.
* p2, 48: Which approach is used to generate the optimization problem solved by GUROBI? The latter is just a general-purpose MILP solver
* p2, l52: What does it mean to "predict the alphabet"?
* p2, l56: Is reference 28 the earliest one in this area?
* p2, l61: Wording unclear, "over a formulation"
* p2, l74: Unclear how minimizing the number of mode switches is a relaxation / simplification of the problem
* p2, l79-84: This paragraph needs to be improved. In particular, no references to existing works using the approach here are given. Are Deep Learning approaches used at all in this area? Furthermore, while at the start of the paragraph ReLUs are mentioned, at the end of the paragraph it is argued that Deep Learning approaches are problematic due to the use of smooth activation functions like sigmoids.
* p2, l87: The technical terms "traces" and "guards" haven't been defined
* p3, l91-92: Unclear, is variational inference used in the EM algorithm or is the EM algorithm regarded as a variational inference technique?
* p3, Definition 1: The formulation is rather unusual and a bit imprecise. For example, $A_1,\ldots,A_m$ are not $m$ different dynamical systems, but rather $m$ matrices inducing $m$ different linear dynamics. Furthermore, the last sentence "We can generalize this to have output maps that also depend on the mode" should be outside the definition.
* p3, l121: The term "observable" has a technical meaning in systems theory, better to use "not observed" here
* p3, l121-122: This sentence is unclear, more details on this approach are needed.
* p3, l135: Typo, "with" instead of "will" needed. Furthermore, the definition of $E_n$ should be moved to the notation section.
* p4, l136: Abbreviation SLS-ID not introduced
* p4, l136-137: Better not to used colons in the sentence
* p4, formula (2): Here $\leq$ seems to be mean entrywise order. Either define this or rewrite the formula in an elementary manner (e.g. $\forall j=1,\ldots,m$ we have $(A_j)_{k,l}\in[-\gamma,\gamma]$ for all $k,l=1,\ldots,n$)
* p4, l148: Is "big M method" a technical term in the field of switched linear systems?
* p4, l149: Abbreviation MILP not introduced
* p4, l150-151: There should be at least one reference for "a common approach"
* p5, l158: This is not really a definition, but rather an algorithm description and should be formatted and labelled accordingly
* p4, l159: Use $:=$ instead of $:$
* p4, l162-164: Are the outputs really the only two cases? This needs to be clarified
* p4, l166: "It is easy to show". At least some details should be provided, maybe in the supplementary material. Furthermore, this statement should be made more precise. For example, what about the case $x_i=0$ and $\|x_i'\|_\infty >> 0$?
* p4, l167: What does "ignoring the technicality related to the limit" mean?
* p5, Algorithm 1, l9: What does $\hat{P}_j$ feasible mean? That this polyhedron is non-empty?
    Furthermore, what happens if this condition is False for all $j=1,\ldots,m$?
* p5, l186: In which space are the polyhedra?
* p7, l300: What is meant by traces? Trajectories?
* p7, l303: Why $N=kT$? If the trajectories have $T$ time steps, then there are only $T-1$ transitions (and each data point correponds to a transition)
* p7, l304: What are "predicted and actual states"? The setting introduced is identification, not prediction
* p8, l314-321: Why was this setting chosen? Is there related work on such tasks? What would be the state of the art?
* p8, l322-327: Details missing. For example, how are the JMM used?
* p9, Figure 2: Better labelling required. In particular, axis labels are missing in both subplots
* p9, l1360-361: It hasn't been explained how the approach can be extended using SARX models

## Supplementary material
* p13, l522-528: This paragraph could be made a bit easier to follow, e.g., to be more explicit.
* p15, Section D: Details on the MILP method and the implementation of the proposed method should be provided here

**Limitations:**

The authors addressed adequately any potential societal or ethical implications.

**Strengths And Weaknesses:**

## 1. Originality
The problem setting seems to be established, but the proposed relaxation approach is new. The algorithm and its analysis are also novel, but based on standard arguments. In particular, in the relaxed problem setting no technical complications appear and no novel techniques seem to be necessary. The algorithm uses a pruning approach based on a treshold of the volume of certain polyhedrons which seems to be new, but due to lack of familiarity with this area this evaluation is not certain.


## 2. Quality
The presentation needs to be improved. The article has language and grammar issues, as well as formatting problems (e.g., using a colon instead of an equality sign). Furthermore, some technical arguments could be explained better, e.g., in the Lemmas in Section 3 or Figure 2. Additionally, there should be more references. Finally, some relevant details on the experiments are missing. There seems to also some potential technical problems, cf. Questions section below.

## 3. Clarity
As already mentioned, the presentation needs to be improved. For example, there is no pseudocode description of the whole algorithm (only some parts of it are presented in Algorithm 1). Some arguments and explanations are also difficult to follow.

## 4. Significance
The identification of switched linear systems is an established and relevant area. The proposed algorithm seems to be working and computationally efficient (in the number of data points, not the state dimension and number of modes). However, the comparison with prior work and state of the art (in particular, in the experimental section) needs to be more thorough, so its difficult to confidently assess the overall significance.

---

> ### Author Response · Authors · 2022-08-02
> **Response to Reviewer FbXu**
>
> We appreciate your thorough reading of the paper and the valuable feedback. Here is our inlined response to your questions and comments.
>
> > How exactly could the SARX model be used to deal with output measurements? Does the algorithm and its analysis need adaptation to this case?
>
> The SARX model simply posits that the next output $y(t+1)$ is a linear function of the previous $n$ outputs: $y(t+1) = f_j(y(t), \ldots, y(t-n+1))$, wherein each $f_j$, $j = 1, \ldots, m$ is an affine map.  With the choice of $m$ (the number of modes) and $n$ (the "look-back") fixed, our algorithm would work with minimal modifications and yield a similar complexity. As a minor caveat, we would be identifying $m \times n$ parameters rather than $m \times n^2$ parameters. To that extent, the complexity would be lower, strictly speaking.
>
> > Page 5, line 203: "then we create $m$ child nodes" This seems incorrect, since in Algorithm 1 a child node $\nu_j$ is only created if $P_j$ is feasible.
>
> You are correct: we did include this bit of detail in line 9 of Algorithm 1. We will add a clarifying comment in the text.
>
> > Proof of Lemma 4, lines 531-532: Does such a leave node (with $\mu=\mu^*$) always exists?
>
> It always exists:  this is proven as part of the proof of Lemma 3. We will modify the statement of Lemma 3 to highlight this. Thank you for noticing.
>
> > Proof of Lemma 5: According to the proof, the lower bound should be […]. How is the lower bound stated in the lemma then derived?
>
> Note that the volume of unit $L_1$ norm ball is $2^n/n!$. Next, we scale this by a factor $\epsilon$ (we will skip the subscript "gap" on $\epsilon$ in this discussion), which gives us the volume $2^n \times \epsilon^n / n!$. Finally, we have a Cartesian product of $n$ such balls (one for each row of the matrix). This gives us volume  $(2^n \times \epsilon^n /n!)^n$ and thus the formula derived in the lemma. In particular, we have $\epsilon^{n^2}$ and not $\epsilon^n$.
>
> > Experiments: Why the choice of $\epsilon_1 = 0$? Does it make the problem easier?
>
>
> The choice of $\epsilon_1 = 0$ yields a version of our algorithm where the "no" answer says that no set of $m$ matrices explains the data with $0$ relative error tolerance. This is just one "natural" default choice we may envision, but others are possible. It does not make the problem any easier, in theory or even in our experience. The complexity bounds we derive depend solely on the gap : $\epsilon_{gap} := \epsilon_2 - \epsilon_1$.
>
> > The MVE is replaced by Chebychev centers. What exactly is lost when using Chebychev centers, i.e., which guarantees cannot be given anymore?
>
> Reviewer Vd1p has a related question and we explain why we did not use semi-definite programming solvers to compute the MVE center.
> It is a common practice while implementing  ellipsoidal methods to replace the MVE center with the Chebyshev center. It provides us with a faster algorithm and is less sensitive to numerical issues.   However, in theory, we do give up the guarantee of Lemma 6 since the Chebyshev center is not the same as the MVE center.
>
> > What is the MILP approach? No details are given
>
> The MILP approach is described up front in Section 2 as a combination of constraints (2), (3) and Lemma 1.
>
> > ​​Page 7, line 303: Why $N=kT$? If the trajectories have $T$ time steps, then there are only $T−1$ transitions (and each data point corresponds to a transition)
>
> It is an indexing issue: by $T$ time steps we mean $T$ transitions starting from the initial states. We will clarify this. Thank you!
>
> > Additional Detailed Remarks
>
> We really appreciate your thorough reading of the paper. We will add appropriate clarifications, and fix the typos you have pointed out. As other reviewers have suggested we will add the pseudocode for algorithm 1. If the paper gets accepted, we will have an extra page to do all that: we think it is quite sufficient.

---

### Official Review · Reviewer_Vd1p · 2022-07-11

**Rating:** 8
**Confidence:** 4
**Soundness:** 4 excellent
**Presentation:** 4 excellent
**Contribution:** 3 good

**Summary:**

The authors study learning switched linear dynamics from transition pairs of states. They exploit a gap in tolerance (over which performance is ill-defined) to guarantee that when a set of feasible matrices becomes too small, no solution is consistent with the constraints. This is used to prune the search tree of a mixed-integer program (MIP). The size of the tree is bounded using a cutting-plane argument when candidate solutions are picked as the centers of maximum volume ellipsoids (within constraint polytopes). The method is tested on synthetic data, author-curated handwriting data, and highway driving data.

**Questions:**

- How are volumes of constraint polytopes computed? Are they computed, estimated, or are nodes simply pruned once the tree reaches depth $K_\min$? This is a minor concern but it should be addressed.
- Why exactly were handwriting data preprocessed to keep points roughly equidistant?
- Should Lemma 9 read $\mathsf{vol}(\widehat{P}_j) \leq (1 - \frac{1}{n})^n \cdot \mathsf{vol}(P_j)$? It seems like the multiplicative factor $(1 - \frac{1}{n})$ should be included for each row of the matrix $A_j$. Additionally, this would sharpen the $n^3$ in Lemma 10 to $n^2$.

**Limitations:**

I believe the authors have adequately captured the limitations of their work and have thought about potential negative societal impact. I think the submission could use a comment on how approaches for switched nonlinear systems may need to be developed (what kinds of approximations could be necessary, localized analysis, etc.), though this is clearly out of the current scope of the work.

**Strengths And Weaknesses:**

# Strengths
- The paper is well-written, the proofs (in the appendix) are easy to follow and the proof summaries provide good baselines from which most of the details can be inferred.
- To my knowledge, the idea is novel and can be extended to several use cases beyond the proposed setting, such as control systems, state-based hybrid systems (with mode transitions at guards), and general mixed integer convex programming.

# Weaknesses
- It is unclear how to specifically prune leaves, as this involves computing the volumes of complex constraint polytopes.
- Some of the design decisions and experimental conclusions were questionable. Specifically:
  - Modern semidefinite programming (SDP) solvers can handle fairly large-scale problems, I would be interested to see how solving for maximum volume ellipsoid centers (as opposed to Chebyshev centers) would affect the performance of the overall approach.
  - The authors claim (in the checklist) that error bars are not informative, but the timing data for TS in Benchmark #2 is couterintuitive (time is not monotonically increasing); this makes me wonder if the behavior is due to a bug, third-party implementation details, randomness, etc., and standard deviations over more trials could help make this behavior more obvious.
  - The decision to space data roughly equally (in Euclidean distance) for the handwriting data set isn't completely justified for me; sometimes real-world systems are affected by dynamics with very different transition behavior throughout the state space. I would be interested in experiments without this preprocessing step.
  - While the predicted velocities for highway data seem good, the "behaviors" extracted from the trajectory modes (in Fig. 2 Right) are not clearly distinguished or obvious to me.

# Typos/Other Issues
- There is a period in the title
- Line 88 and 324, Markov should be capitalized
- The sentence on lines 101-102 is confusing
- Line 137 should read "and *an* absolute limit $\gamma$"
- For Lemma 1 to be necessary and sufficient as worded, the matrices should explain the data away with the mentioned error bound and tolerance, but also the absolute bound $\gamma$
- Line 160, data should be lowercase
- Page 6 was the first place I noticed nonstandard use of colons (to indicate setting a variable?)
- I believe the cited proposition on lines 260-261 should be Proposition 3.7.1
- Line 273, Cartesian should be capitalized
- Line 336-337 should read "which in *turn* can be useful"
- Line 348 should read "turning for changing *lanes*"
- Line 495 should read "*computing* convex sets"
- I believe the citation in Line 508 should be Section 8.4.2
- Should Line 553 (also 592) say that the path from the root to the leaf is of length *at least* $mK$? This seems to agree with the use of $T$ later (which is not introduced)
- Line 569 should read "as in *the* proof of Lemma 7"
- The proof of Lemma 10 is repetitive after the proof of Lemma 7, consider modifying this to highlight only the differences
- Line 617, "linear line" is redundant
- Line 619, I believe $i$ should take values from 0 to $T_{\mathrm{raw}} - 1$
- Page 17, the figure is hard to read. Moreover, the weights on each edge are all listed as 1.0

---

> ### Author Response · Authors · 2022-08-02
> **Response to Reviewer Vd1p**
>
> Thank you for your insightful comments and feedback. Our responses to your questions are inlined below.
>
> > It is unclear how to specifically prune leaves, as this involves computing the volumes of complex constraint polytopes.
>
> Actually, we can directly use Lemma 4: a node is a leaf if it no longer contains a norm-ball of sufficient size. This is easy to check and __involves no volume computation__, since we can use the same LP formulation as computing the Chebyshev center to check this. The volume shrinking argument in our paper assures us that we will reach such a leaf within a bounded depth.
>
> > Modern semidefinite programming (SDP) solvers can handle fairly large-scale problems, I would be interested to see how solving for maximum volume ellipsoid centers (as opposed to Chebyshev centers) would affect the performance of the overall approach.
>
> We agree with you: however, in our defense we have been working with SDP solvers, and have dealt with numerical issues that are mostly not present in LP solvers. In our past experience, MVE computation often fails when the polyhedron is "highly skewed" – we sometimes end up with numerical issues that push the point outside the polyhedron. Nevertheless, we have not tried MVE computation for this particular algorithm and completely agree with the need for a detailed comparison: we will add it to our subsequent version.
>
> > The authors claim (in the checklist) that error bars are not informative, but the timing data for TS in Benchmark #2 is counterintuitive (time is not monotonically increasing); this makes me wonder if the behavior is due to a bug, third-party implementation details, randomness, etc., and standard deviations over more trials could help make this behavior more obvious.
>
> Benchmark # 2 timing is due to the particular synthetic dataset. With N = 10000 data points, it just so happens that  the matrices explaining away the data are found rapidly by our algorithm while performing the tree exploration, whereas it takes more time for N = 1000.   To provide a better overall evaluation, we will generate and report on 10 different synthetic benchmarks/data-sets for each size/number of modes (including the error bar).
>
> > The decision to space data roughly equally (in Euclidean distance) for the handwriting data set isn't completely justified for me;
>
> The short answer is that it depends on what we would like to model. In our case, we were interested in modeling alphabet shape.
>
> Without the pre-processing, however, we would be capturing not just the shape of the alphabet but (a) artifacts of how people draw alphabets including where they go faster and where they go slower (interesting); (b) artifacts of how the software samples positions from the touch-pad (maybe not so interesting). For the application at hand, our approach can be implemented online by implementing a filter on top of the raw data stream to output interpolated samples that are roughly equidistant (but these samples would not be regularly spaced in time). We think this way of filtering the data is best for modeling/predicting the shape but agree that it may not be the best approach to model other aspects of human handwriting on a surface.  For those aspects, we could  explicitly  include time difference $dt$ to the dynamics, e.g., $x_{t+1} = f(x_t, dt)$ and also estimate the  writing velocity as a state variable. This would make the model able to capture writing speed better.
>
>
> > While the predicted velocities for highway data seem good, the "behaviors" extracted from the trajectory modes (in Fig. 2 Right) are not clearly distinguished or obvious to me.
>
> The figure is not well explained and we will fix it. The "gray" mode has the effect of making the lateral  velocity (along the x direction) more positive (i.e, car moves to the right) whereas the "light-blue" mode applies feedback that makes the lateral velocity more negative (i.e, car moves to the left). However, the confusion arises since there are cases when the lateral velocity is small (the car is moving straight) that both modes have very similar behavior. In these cases, what is labeled as a blue mode could very well be a gray mode since either mode can explain away the data.
>
> >Should Lemma 9 read $vol(\hat{P}_j) \leq (1−1/n)^n vol(P_j)$? It seems like the multiplicative factor $(1−1/n)$ should be included for each row of the matrix $A_j$. Additionally, this would sharpen the $n^3$ in Lemma 10 to $n^2$.
>
>
> Note that the reviewer seems to implicitly assume that the addition of new constraints will “invalidate” each and every row of $Q_j$ but this is not always the case. All we can guarantee is that at least one row will be invalidated. The term $(1 - 1/n)$ should not be raised to the $n$th power. Thus, the refined analysis in Appendix C is the best we can do.

---

> > ### Comment · Reviewer_Vd1p · 2022-08-05
> > **Author responses are sufficient**
> >
> > The authors have answered my questions and addressed my concerns. I believe subsequent versions will benefit if the authors add parts of the explanations from their responses.
> >
> > The writing (specifically lines 253-254) makes it seem like polyhedral volume computation is used as a stopping condition. If norm ball volume computation from Lemma 4 should be used instead, I think this should be made clearer in the exposition.
> >
> > I see how I misunderstood the reason for the multiplicative factor in Lemma 9, I believe the authors are correct.
> >
> > In the alphabet data set, I agree with the authors that artifacts due to the touch pad are not relevant for modeling. I am satisfied with the authors' justification, but I still contend that writing speed is an important factor (perhaps this warrants a sentence or two).
> >
> > I appreciate the explanation of the modes in the driving data (and agree that such an explanation should be included). The particular examples chosen do not make it obviously clear that the modes are what the authors say they are, but this is minor to me.

---

### Official Review · Reviewer_3C4e · 2022-07-18

**Rating:** 5
**Confidence:** 4
**Soundness:** 2 fair
**Presentation:** 2 fair
**Contribution:** 1 poor

**Summary:**

This paper presents an algorithm for learning switched linear dynamical systems. An algorithm is proposed which is linear in the size of the problem and the proposed algorithm is demonstrated on some benchmark systems.

**Questions:**

Please try and answer these:
1.Switched systems can also be modeled using complementarity constraints. It would be good to compare or contrast against those techniques. Do you think that your method could be more efficient than the methods using complementarity constraints? For example, see Contactnets by Post et al. or https://arxiv.org/pdf/2203.10013.pdf.
2. The proposed method has been shown for autonomous systems only. Would it work for controlled systems, e.g. where x_k+1=Ax_k+Bu_k?
3. One the practical problems of estimating or learning in hybrid systems is dealing with imbalance in data between different modes? To make any of these algorithms useful for, say robotic systems, that is a problem that needs to be solved. How do you suggest your algorithm will work if there is an imbalance in data between different modes?
4. Have you tested you method when there are lots of possible modes in the system? A lot of practical systems like robotic systems usually have a lot of modes that can arise? How does your method scale with the number of modes? Please explain.


**Ethics Review Area:**

["I don’t know"]

**Limitations:**

Some limitations have been described. I would like authors to think about the limitations I pointed out.

**Strengths And Weaknesses:**

Strengths:
1. the problem is well motivated, and still remains open in so many aspects.
2. The analysis of the proposed method is interesting.
Weaknesses:
1. the comparison of the proposed work is weak and makes evaluation of the proposed work difficult.
2. Switched systems can also be modeled using complementarity constraints. It would be good to compare or contrast against those techniques.
3. The proposed method has been shown for autonomous systems only. Would it work for controlled systems, e.g. where x_k+1=Ax_k+Bu_k?

---

> ### Author Response · Authors · 2022-08-02
> **Author Response to Reviewer 3C4e**
>
> Thank you for your insightful comments: we appreciate your feedback. Our response to your comments are inlined below.
>
> > The proposed method has been shown for autonomous systems only. Would it work for controlled systems, e.g. where $x_{k+1}=A x_k+B u_k$?
>
> This is a very good point. The proposed method will work for identifying a finite number ($m$) of affine maps $f_j: R^n \rightarrow R^p$ for any $n, p \geq 1$. Thus, if the control inputs $u_k$ are given then the method __easily extends__ to this case. This is similar to our highway driving example, wherein we have exogenous control inputs and our aim is to learn feedback functions  $u = K_j x$, where we assume $m$ different feedback laws.
>
> However, if the control inputs $u_k$ are not given, the problem is much harder since we will end up with bilinear constraints involving unknown $B$ and $u_k$.
>
> > Switched systems can also be modeled using complementarity constraints. It would be good to compare or contrast against those techniques. Do you think that your method could be more efficient than the methods using complementarity constraints? For example, see Contactnets by Post et al. or https://arxiv.org/pdf/2203.10013.pdf.
>
> This is an interesting idea that we have not yet considered. On one level, based on the paper you mention, we could formulate linear programming with complementarity constraints (LPCCs) instead of MILPs for solving the identification problem. It is an interesting alternative to the MILP formulation but to the best of our knowledge LPCCs remain NP-hard to solve and this approach does not avoid the exponential dependence on the number of data points. Your comment also suggests a different modeling approach to begin with, where we would formulate a hybrid systems with large number of modes using a small number of complementarity constraints (used to model contact forces in a robotic system).  We will definitely discuss this in our related work.
>
> It does give us some interesting ideas for future work – thank you!
>
> > One of the practical problems of estimating or learning in hybrid systems is dealing with imbalance in data between different modes? To make any of these algorithms useful for, say robotic systems, that is a problem that needs to be solved. How do you suggest your algorithm will work if there is an imbalance in data between different modes?
>
> Note that we make no assumptions about how the data are distributed across modes, or alternatively, what kind of switching sequences we encounter. For instance, it could be possible that one of the modes has very few data points corresponding to it (e.g., a single data point). Our method would still apply. Obviously, if one of the modes has very little data we will identify some matrix for that mode that may be different from the underlying ground truth since we will have an under-constrained problem for that mode.
>
> > Have you tested your method when there are lots of possible modes in the system? A lot of practical systems like robotic systems usually have a lot of modes that can arise? How does your method scale with the number of modes? Please explain.
>
> The dependence on modes is twofold: (1) the branching for assigning a mode to each data point, and (2) the number of unknowns, equal to $d := m \times n^2$. This is reflected in the overall complexity: $O(m^{m \times n^3})$.  The complexity is thus exponential and we have tried it examples with up to $8$ modes.
>
> However, if we have many modes, but prior knowledge on the matrices, then the number of unknowns is  smaller than $m \times n^2$: $d \ll m \times n^2$, so that the overall complexity is still tractable for that kind of problem with a large number of modes. As an illustrative example: we could assume that each row in the matrix can have $3$ possibilities and we have $10$ such rows. We would use the algorithm presented in this paper individually for each row, where it would be quite fast. However, that means that we are identifying $3^{10}$ possible modes  in a global sense! One of our future plans is to identify how we may parameterize such “prior knowledge” succinctly in order to identify systems with a large number of modes efficiently.

---

> ### Comment · Reviewer_3C4e · 2022-08-07
> **Post rebuttal comment**
>
> I would like to thank the authors for their reply.
>
> -- I understand that the complementarity-based approach is another alternative to what the authors have provided but I think such a comparison is important for the completeness of the paper. Complementarity-based modeling of switched systems is a well known approach which avoids mode enumeration and thus, might be more efficient.
> -- The above discussion is also applicable to their compute complexity. It is well known that if we have to enumerate all modes, the complexity will be exponential.
> -- I think the paper has limited applications under the current assumptions.

---

> > ### Author Response · Authors · 2022-08-08
> > **Response to post rebuttal comment**
> >
> > Thank you for the comment.  We agree that the technique _as presented in the paper_, will not efficiently address the application you have in mind to modeling robotic systems with contact forces. This is because the technique in the paper makes no assumptions on the matrices for the various modes. The problem becomes easier if such assumptions can be made.
> >
> > For instance, in the "PyRoboCOP" paper that you cite (https://arxiv.org/pdf/2203.10013.pdf) one of the examples (cart-pole with soft walls, Fig. 3) has multiple modes but the dynamics of the mode share parameters such as the spring constants $K_1, K_2$ and the mass of the pole $m_p$. Our method as presented in the paper is not designed to exploit it. However,  if modified appropriately, it can perform a tree exploration by assigning modes to various data points. But in doing so it can "zero in" on values of  $K_1, K_2, m_p$ that are accurate enough to meet our tolerance constraints. When this happens the rest of the algorithm will finish rapidly since the parameters identified on the few modes through the initial tree exploration will rapidly "fill in the blanks" for the other modes. Thus, it is not exactly true that the approach presented in the paper will always need to do a full mode enumeration.
> >
> > The key here is that you can have a large number of modes but if the matrices of these modes are based on affine combinations of some shared parameters, we can adapt the method of this paper to exploit that sharing.
> >
> >
> >
> > Another difference between the two approaches is that complementary systems involve state-based switching, and are in this sense closer to piecewise affine systems and other similar hybrid models (https://www.sciencedirect.com/science/article/pii/S0005109801000590), while the systems considered in the paper (for the theory and applications) involve arbitrary switching (in the car application: the behavior of the car results from driver's decisions often independent from the state of the car; while in the handwriting application: the letters are made of "primitives" combined in an non-deterministic way when considered over the whole alphabet). The arbitrary switching feature of the systems that we consider prevents us from using complementary systems learning techniques in our applications. On the other hand, our approach (for the moment) cannot identify  state-based switching rules: we focus simply on learning the dynamics for each mode.
> >
> > Based on the ideas in this paper, we have plans to tackle more applications in the future that could be of interest for many communities that study this problem.
> >
> > Finally, we can facilitate comparison with complementarity-based modeling by trying to tackle the cart-pole example with our approach as is. But that example does not have too many modes in our estimation, and we expect our approach to work well.
> >
> > Thanks again!

---

> > > ### Comment · Reviewer_3C4e · 2022-08-09
> > > **update**
> > >
> > > Thanks for the answers. I have updated and submitted my final comments.

---

### Meta-Review · Area_Chair_KtDf · 2022-08-26

**Recommendation:** Accept
**Confidence:** Certain

**Metareview:**

This was a borderline paper and after discussion with the reviewers, we have decided to accept the paper. The paper currently has many typos and the approach requires more comparison with other approaches for learning switched systems--- though these can be addressed in the final revision. Overall, the paper address a relatively open question from a machine learning viewpoint and the results are interesting and the approach is novel.

**Award:**

No

---

### Decision · Program_Chairs · 2022-09-14

Accept